# Land Use Transition and Driving Forces in Chinese Loess Plateau: A Case Study from Pu County, Shanxi Province

Han Huang [1,2,3], Yang Zhou [1,2,3,*] , Mingjie Qian [4] and Zhaoqi Zeng [1,3]

1   Center for Assessment and Research on Targeted Poverty Alleviation, Institute of Geographic Sciences and Natural Resources Research, Chinese Academy of Sciences, Beijing 100101, China; huanghandida2018@163.com (H.H.); zengzhaoqi24@icloud.com (Z.Z.)
2   Key Laboratory of Regional Sustainable Development Modeling, Chinese Academy of Sciences, Beijing 100101, China
3   College of Resources and Environment, University of Chinese Academy of Sciences, Beijing 100049, China
4   School of Land Science and Technology, China University of Geosciences, Beijing 100083, China; qianmingjie@cugb.edu.cn
*   Correspondence: zhouyang@igsnrr.ac.cn; Tel.: +86-10-6488-9034

**Abstract:** Land use transition is essentially one of the manifestations of land use/cover change (LUCC). Although a large number of studies have focused on land use transitions on the macro scale, there are few studies on the micro scale. Based on the data of two high-resolution land use surveys, this study used a land use transfer matrix and GeoDetector model to explore the spatial-temporal patterns and driving forces of land use transitions at the village level in Pu County over a ten-year period. Results show that Pu County has experienced a drastic process of land use transition. More than 80% of cropland and grassland have been converted to forest land, and over 90% of the expansion of built-up land came from the occupation of forest land, cropland, and grassland. The driving forces of land use transition and its magnitude depended on the type of land use. The implementation of the policy of returning farmland to forest, or grain-for-green (GFG) was the main driving force for the large-scale conversion of cultivated land to forest land in Pu County. In the context of policy of returning farmland to forests, the hilly and gully regions of China's Loess Plateau must balance between protecting the ecology and ensuring food security. Promoting the comprehensive consolidation of gully land and developing modern agriculture may be an important way to achieve a win-win goal of ecological protection and food security.

**Keywords:** land use transition; spatiotemporal pattern; driving forces; GeoDetector model; Chinese Loess Plateau

## 1. Introduction

Over the past 40 years, China has experienced an unprecedented process of industrialization and urbanization. With rapid economic growth, China's land use-related problems, such as dramatic built-up land expansion, high-quality arable land loss, land degradation, soil pollution and inefficient rural homestead use, have emerged simultaneously [1–4]. From 1998 to 2018, China's annual growth rate of urbanization reached 1.04%, and more than 600 million rural population flowed to cities. Population mobility has also aggravated the contradiction of built-up land in urban and rural areas [5,6]. To some extent, the problem of unsustainable land use has promoted research on the optimal allocation of land use, the coordination of sustainable economic development, and the protection of cultivated land, as well as the process, pattern, and effect measurement of land use transformation [7–9].

The groundbreaking research on land use transformation originated in the 1990s, and was based on the assumption of forest land transition [10,11]. In the following decades, researchers have been concerned aboutforest land transition [12]. Numerous empirical

studies emphasized specific countries and regions, e.g., South Korea, Bhutan, Vietnam, and Australia, as well as Mississippi (USA) and other Southeast Asian countries, etc. [13–17]. The driving force of land use transformation has also received extensive attention from different perspectives, such as economic development, social transition, ecological protection, and land management [18–20]. At the beginning of the 21st century, the concept of land use transformation was introduced into China, and then attracted the extensive attention of Chinese scholars [21]. The connotation of land use transformation has been enriched and developed. Previous studies have focused on the process, pattern and driving mechanisms of land use transformations in China and typical regions [22–25]. The theory of land use transition has been widely used in research on land use-related issues, such as village hollowing, farmland conversion, and construction land expansion, which greatly promotes the vigorous development of land science and rural geography [1,8,9,26].

China's Loess Plateau is a very vulnerable region, with the most serious soil erosion in the world [27,28]. Due to a special soil-forming environment, extreme drought climate conditions, and strong human activity intervention, its land use has undergone significant changes over the past decades [29]. The fragmentation of the terrain and the ravines of the hills and valleys in the region have led to low agricultural production potential, which has severely restricted the flow of the means of production and the improvement of production technology [30]. There is a vicious circle between ecological degradation and the transition of resources [30,31]. Since 1990, the Chinese government has implemented a series of ecological protection policies, which have greatly improved the ecology of the Loess Plateau [31]. Early research on land use change in the Losses Plateau focused on the macro-medium level, but the micro-level research was relatively insufficient [28,32–34]. To fill this gap, based on the high-resolution and high-quality land use survey data of Pu County, Shanxi Province in 2009 and 2019, this study applied a land use transfer matrix and GeoDetector model to investigate the spatial-temporal pattern and evolution mechanisms of land use transformation at the village scale in typical areas of the Loess Plateau. Our findings would be of practical reference for local policymakers to formulate land use policies and promote the national strategy of rural revitalization.

## 2. Materials and Methods

### 2.1. Study Area

Pu County (N 36°11′32″~36°38′13″, E 110°51′09″~111°23′36″) is located in the southwest of Shanxi Province and the hinterland of the Loess Plateau gully region (Figure 1). Pu County has 9 townships and 98 administrative villages, with a land area of 1512.9 square kilometers. Surrounded by mountains in the east, south, and north, the terrain of Pu County is fragmented, with a distribution characteristic of highland in the east and lowland in the west. The Xinshui River, a tributary of the Yellow River, winds through Pu County from east to west for 70 km. In 2019, the county had a population of 0.112 million, of which its urban and rural population accounted for 49.44% and 50.56%, respectively.

There were three main reasons for choosing Pu County as the study area. Firstly, Pu County is a typical county on the Loess Plateau gully region in China. As one of the most severely ecologically fragile areas in the world, China's Loess Plateau has suffered severe soil erosion and water shortages [31]. Secondly, Pu County is a microcosm of the most under-developed areas in China, and has been experiencing rapid urbanization and economic growth. More than 30,000 rural people in Pu County have flowed into cities for better jobs and livelihoods, and its per capita GDP in 2019 was five times higher than that in 2009. Thirdly, a series of policies and measures, such as the grain-for-green (GFG) policy and land consolidation projects, have been implemented in Pu County. Statistics show that the county has implemented 37 land consolidation projects (8839.68 ha) from 2016 to 2020, supplementing 1273 ha of arable land. In view of Pu County's natural resources and geographical environment conditions, as well as human intervention, it was therefore selected as our research area.

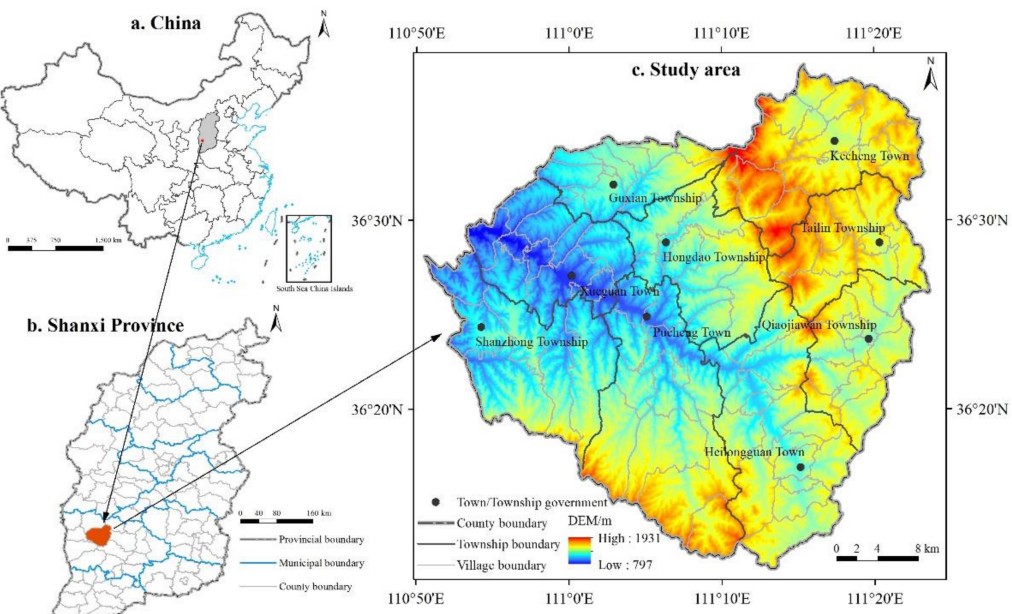

**Figure 1.** Location of Pu County (**a**), and of Shanxi Province (**b**) in China (**c**).

*2.2. Data Source*

The data used in this study include vector data (land use, traffic networks, village boundaries) and raster data (topography, GDP and population) in Pu County from 2009 to 2019. The vector data of land use was obtained through the Second National Land Survey and the Third National Land Survey taken in 2009 and 2019 by Land and Resources Administration Departments in Pu County. Due to different classification standards of land use in two phases, a scientific and practical classification method was applied to reclassify the original second-class land use types and merged into seven first-class types according to land use planning, which includes cropland, orchard, forest land, grassland, built-up land, water bodies, and unused land. Compared with crop farming, the development of the forestry and fruit industry has good socioeconomic and eco-environmental benefits [35]. The data on provincial, county and village boundaries, central location of township and county government were provided by the Pu County Natural Resources Bureau. Digital elevation model (DEM) with a spatial resolution of 90 m and GDP and population data with one-kilometer grid (KMG) were available from the Resource and Environment Data Cloud Platform of Chinese Academy of Sciences (CAS). Traffic vector data in Pu County were obtained from the National Earth System Science Data Center of CAS (http://www.geodata.cn).

*2.3. Methods*

2.3.1. Transfer Matrix of Land Use

The Markov transfer matrix was derived from the quantitative description of the state and state transition over a certain time period. Based on the vector data, the dynamic information of land use was measured by transfer matrix at a certain period of time. Then, the land use transfer rate and change intensity between different land use types was calculated by the following formula:

$$S_{ij} = \begin{bmatrix} S_{11} & S_{12} & \dots & S_{1n} \\ S_{21} & S_{22} & \dots & S_{2n} \\ \dots & \dots & \dots & \dots \\ S_{n1} & S_{n2} & \dots & S_{nn} \end{bmatrix}, \quad D_i = \sum_{j=1}^{n} S_{ij} - S_{ii}, \quad D_j = \sum_{i=1}^{n} S_{ij} - S_{jj} \quad (1)$$

where $S_{ij}$ is the area converted from land-use type $i$ to type $j$; $D_i$ and $D_j$ represent the reduced area of land-use type $i$ and the increased area of type $j$ during the study period, respectively.

### 2.3.2. Variable Selection and Parameterization

Land use transition refers to the process of regional land use changing from one form to another, corresponding to the transformation of economic and social development stages in a period of time [12]. Limited by the availability of data at the village level, the area of land use change was selected as the dependent variable, and nine factors including terrain, slope, road density and distance from river were selected as the independent variables to explore the driving forces of land use transformations (Table 1).

**Table 1.** Driving factors and their codes, definitions, and units.

| Type | Factor | Code | Definition | Unit |
|---|---|---|---|---|
| Natural factors | Elevation | X1 | The average slope in a village | m |
| | Slope | X2 | The average slope in a village | |
| | Relief | X3 | The average relief degree of land surface | m |
| | Distance to river | X4 | The total change of linear distance from village to rivers | km |
| Traffic factors | Road density | X5 | The total change of road density in a village between 2009 and 2019 | km/km$^2$ |
| Locational factors | Distance to county | X6 | The linear distance from village to the county government | km |
| | Distance to town | X7 | The linear distance from village to the town/township government | km |
| Social and economic factors | GDP | X8 | The total change of GDP per unit in a village between 2009 and 2019 | 10,000 yuan/km$^2$ |
| | Population | X9 | The total change of population per unit in a village between 2009 and 2019 | people/km$^2$ |

**Natural factors**: Geographical environment is the first geographical factor, including terrain, hydrology, climate, soil, and vegetation, etc. Restricted by differences in spatial scales, the applications of vegetation, climate, and soil are mostly large-scale or medium-scale, and the spatial differences are not significant in the small-scale range. This research used villages as the research unit; therefore, the research scale was relatively small. We chose altitude, slope, surface undulation and the distance between the village and the nearest river to characterize the geographical environment conditions.

**Traffic conditions**: The traffic network is an indispensable passage offering a connection between the interior and exterior areas of regions, and it refers to interconnected and network-distributed road systems at various levels for traffic requirements in a certain area. The more developed the traffic network is, the closer the external exchanges are, and the higher the degree of land use will be, relatively. Hence, we chose the road density to reflect the traffic conditions in Pu County.

**Geographical location**: Location condition is usually characterized by the linear distance to local governments [36,37]. Generally, land use changes around county centers and surrounding towns with better geographical conditions are more intense. This study took the distance from the villages to the center of the county and township government as the proxy variable of geographic location.

**Socioeconomic development**: Population is one of the most active elements in both urban and rural areas. Massive migration of population and rapid accumulation of capital will inevitably lead to development or idleness of a large amount of urban–rural land use, in which land use patterns will be changed. On the basis of this, the variation of GDP and population per unit were chosen as the proxy variables of socioeconomic development.

### 2.3.3. GeoDetector Model

GeoDetector is a statistical model to detect the spatial heterogeneity of geographical objects [38]. In this study, GeoDetector was used to detect the degree to which driving

factors explained the spatial heterogeneity of land-use changes, and its *q*-value was defined as follows:

$$q = 1 - \frac{1}{n\sigma^2} \sum_{i=1}^{m} n_i \sigma_i^2 \qquad (2)$$

where *i* = 1, . . . , *m* represents the strata of driving factors, $n_i$ and *n* denote the number of villages in the strata *i* and in the entire area, respectively, and $\sigma^2$ and $\sigma_i^2$ are the variance of transition area of driving factors in the strata *i* and in the entire area, respectively. The *q*-statistic denotes the driving forces of the factor on transition area, and its value exists between 0 and 1. A larger *q*-value indicates stronger forces of determination to land use transitions. *q* = 0 if a factor is totally irrelevant to transition, and *q* = 1 if the transition is completely controlled by one factor. In this study, the *q*-statistics of nine factors in six types of land use transformation were calculated.

## 3. Results

### 3.1. Spatiotemporal Pattern of Land Use Transition in Pu County

3.1.1. Quantitative Changes of Land Use During the Period 2009–2019

Table 2 shows the changes of seven land use types in transfer-in areas (TIAs) and transfer-out areas (TOAs) of Pu County from 2009 to 2019. We also introduced the total change area (TCA), net change area (NCA), and exchange change area (ECA) to further calculate the absolute value of the difference between TIA and TOA, and the difference between TCA and NCA, respectively. Results showed that forest land had experienced the largest TCA (281.384 km$^2$), followed by grassland (179.622 km$^2$) and cropland (167.644 km$^2$). The TCAs of built-up land, orchard, unused land, and water bodies were relatively small. The NCAs of forest land, grassland and unused land were much higher than their ECAs, which denoted that changes of these land use types were mainly manifested as the net changes in areas of inflow and outflow over the study period. On the contrary, the ECAs of built-up land, water bodies and orchard accounted for a higher proportion of TCAs compared with their NCAs. The total change areas of built-up land, water bodies, and garden plots in the study area were not obvious. The TCA of cropland reached 167.644 km$^2$.

**Table 2.** Land use changes in Pu County over the study period (km$^2$).

| Land Use Type | TIA | TOA | TCA | NCA | ECA |
|---|---|---|---|---|---|
| Cropland (CL) | 48.254 | 119.390 | 167.644 | 71.136 | 96.508 |
| Orchard (OR) | 6.494 | 4.145 | 10.639 | 2.349 | 8.290 |
| Forest land (FL) | 250.299 | 31.085 | 281.384 | 219.214 | 62.170 |
| Grassland (GL) | 12.112 | 167.51 | 179.622 | 155.398 | 24.224 |
| Built-up land (BL) | 21.319 | 11.880 | 33.199 | 9.439 | 23.760 |
| Water bodies (WB) | 3.302 | 1.331 | 4.633 | 1.971 | 2.662 |
| Unused land (UL) | 0.046 | 6.484 | 6.530 | 6.438 | 0.092 |
| Total | 341.825 | 341.825 | 341.825 | 232.972 | 108.853 |

Figure 2 shows the change range of each land use type in Pu County. Results demonstrate that, over the study period, the areas of orchard, forest land, built-up land, and water bodies increased by 22.66%, 29.24%, 22.54%, and 42.82%, respectively, while cropland, grassland, and unused land increased by 26.04%, 36.45%, and 95.01%, respectively. Compared with 2009, the unused land in Pu County decreased by 95.01% in 2019, reflecting the improvement of land use efficiency to a certain extent. The large-scale decrease in cultivated land and the substantial increase in forest land were mainly driven by the implementation of GFG and ecological protection policies in Pu County. Driven by economic interests, grassland in some areas has been used to develop forestry and fruit industry. Meanwhile, affected by economic development and population growth, the built-up land of Pu County has expanded by 22.54% in the past 10 years.

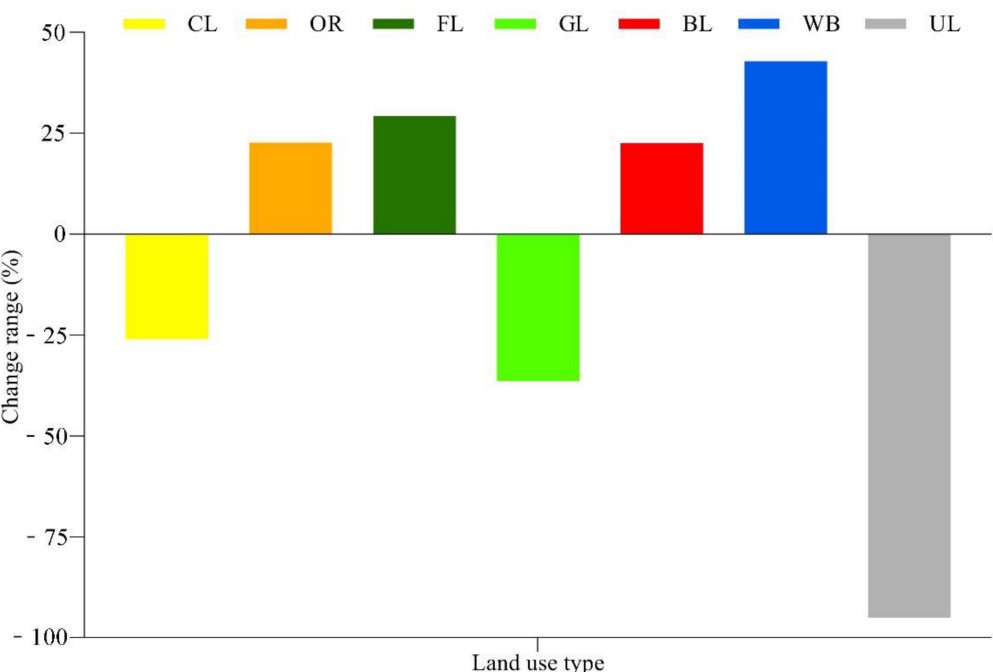

**Figure 2.** Land use changes in Pu County between 2009 and 2019. Notes: CL, OR, FL, GL, BL, WB, and UL represent cropland, orchard, forest land, grassland, built-up land, water bodies, and unused land, respectively.

Table 3 displays the land use transfer matrix in Pu County between 2009 and 2019. Results show that Pu County's land use pattern has experienced significant changes, which were mainly characterized by the transformation of cropland to forest land and built-up land, forest land to cropland and built-up land, and grassland to cropland and forest land, with the conversion areas of 103.3 km$^2$, 7.5 km$^2$, 18.4 km$^2$, 7.7 km$^2$, 21.9 km$^2$ and 139.1 km$^2$, respectively. Land use changes can be regarded as a result of several contributions of other types of land use by the transfer matrix; this approach is one of the most common ways to explore the internal conversions of land use between different land use types (Figure 3). From the perspective of internal conversion, the decrease in cropland and grassland was mainly due to the expansion of built-up land, and the reduced areas accounted for 86.6% and 83.0% of transfer-out areas of cropland and grassland, respectively. Besides, cropland and grassland near flood plains were gradually reclaimed for river courses by soil erosion harness, and these two types contributed to 41.6% and 26.8% of the transfer-in area of water bodies, respectively. Moreover, unused land was mainly converted into grassland and woodland, which may be due to the cultivation of meadow and shrubs considering ecological protection. Additionally, from 2009 to 2019, built-up land expanded by 9.4 km$^2$, largely because of transitions from forest land and cropland, which accounted for 36.3% and 35.3% of the transfer-in area of built-up land, respectively. In the context of rapid urbanization and rural revitalization, cultivated land and forest land in the urban–rural junctive region was occupied or requisitioned for new housing with the expansion of built-up areas. As roads were built or widened, massive areas of forest land, afforested cover, might be transformed into built-up land for transportation.

3.1.2. Changes of Land Use Structure During the Period 2009–2019

Figure 4 displays the changes of land use structure over the whole period 2009–2019 in Pu County. Results indicates that forest land, grassland, and cropland were the main land use types in Pu County in 2009 and 2019, and the sum areas accounted for more than 95% of the total areas, while the proportions of other types were less than 5%. The proportions of three main land use types in 2009 were 49.55% for forest land, 28.18% for grassland, and 18.06% for cropland, and the ratios of these three land use types in

2019 reached 64.04%, 17.91%, and 13.35%, respectively. From 2009 to 2019, under the background of rapid transformations from cropland and grassland, forest land expanded continuously. Compared with cropland and grassland, planting trees not only enables farmers to obtain higher economic benefits, but also plays a more positive role in curbing soil erosion and protecting ecological environment, which drives the transformation of grassland or cropland into forest land. Faced with increasing pressure on rapid population growth and food security, the declining rate of cultivated land area was slower than that of grassland. The areas of grassland loss accounted for 10.27% of the total area while that of cultivated land were only 4.71%. Actually, the effective implementation of National Programs for Food Security has been accelerating the transition from traditional agriculture to modern agriculture. Over the study period, the proportion of built-up land increased slightly from 2.77% to 3.39%. There is no doubt that the construction of housing, the expansion of the road network, and the improvement of infrastructure will inevitably occupy land (mainly arable land and unused land), which is particularly prominent in areas with rapid economic growth.

**Table 3.** Transfer matrix of land use in Pu County between 2009 and 2019 (km$^2$).

| Land Use Type in 2009 | Land Use Type in 2019 | | | | | | | |
|---|---|---|---|---|---|---|---|---|
| | CL | OR | FL | GL | BL | WB | UL | Total |
| CL | 153.766 | 5.112 | 103.326 | 2.534 | 7.528 | 0.884 | 0.006 | 273.156 |
| OR | 2.741 | 6.222 | 0.631 | 0.441 | 0.326 | 0.006 | 0.000 | 10.367 |
| FL | 18.369 | 0.509 | 718.604 | 3.987 | 7.743 | 0.465 | 0.011 | 749.689 |
| GL | 21.912 | 0.768 | 139.090 | 258.874 | 4.342 | 1.374 | 0.022 | 426.384 |
| BL | 3.481 | 0.102 | 5.177 | 2.706 | 30.005 | 0.409 | 0.005 | 41.885 |
| WB | 0.596 | 0.002 | 0.366 | 0.170 | 0.197 | 3.272 | 0.000 | 4.603 |
| UL | 1.153 | 0.001 | 1.709 | 2.272 | 1.184 | 0.164 | 0.292 | 6.776 |
| Total | 202.020 | 12.716 | 968.903 | 270.986 | 51.324 | 6.574 | 0.338 | 1512.860 |

Notes: CL, OR, FL, GL, BL, WB, and UL represent cropland, orchard, forest land, grassland, built-up land, water bodies, and unused land, respectively.

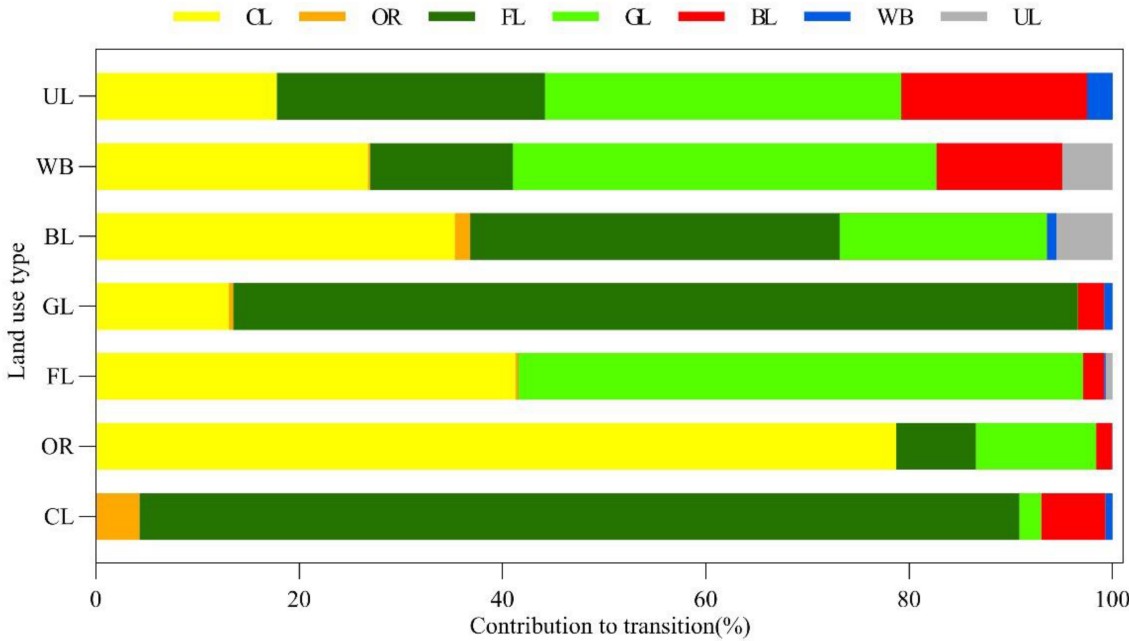

**Figure 3.** Internal conversions between land use types and contributions to land use transition. Notes: CL, OR, FL, GL, BL, WB, and UL represent cropland, orchard, forest land, grassland, built-up land, water bodies, and unused land, respectively.

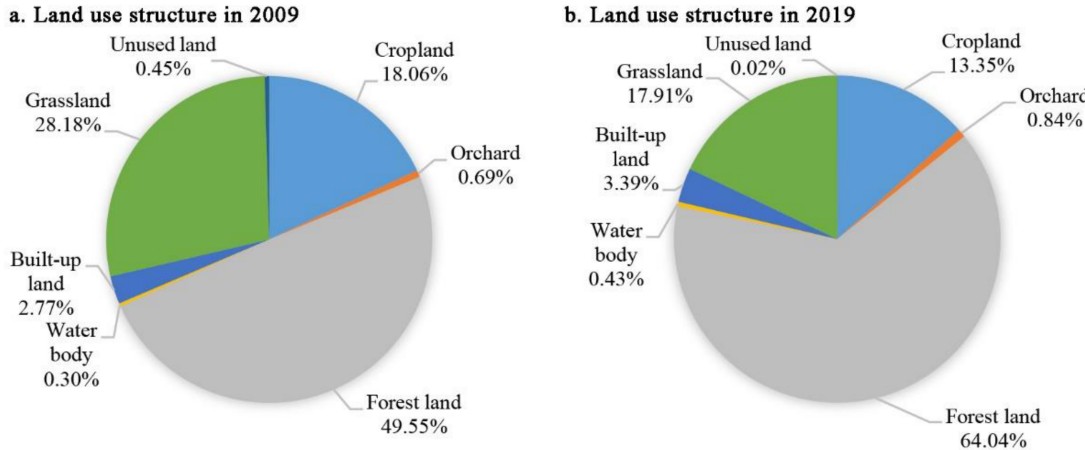

**Figure 4.** Changes of land use structure in Pu County between 2009 and 2019.

### 3.1.3. Changes of Typical Land Use Types in Pu County

Based on the vector land use data, land-use maps in Pu County in 2009 and 2019 were drawn (Figure 5). It was obvious from the two maps that forest land, grassland, and cropland were the main land use types over the study period. In terms of the spatial distribution of land use patterns, forest land was the most widely distributed over the study area, especially in the east and south of Pu County at higher altitudes. On the contrary, grassland was largely located in the middle and northwest regions, with a lower terrain. Generally speaking, built-up land was mainly distributed in the suburbs, designated towns, rural settlements and traffic networks at all levels. The overall distribution of built-up land in Pu County was sporadic, in which rural settlements were distributed in dots or blocks along the main roads. For water bodies, the areas accounted for a small proportion and were mainly distributed in strips along the gully and low-lying areas of the county. Among them, the Xinshui River and its tributaries that flow through Xueguan Town, Pucheng Town, and Heilongguan Town from west to east are the main water resources of Pu County. The overall distribution of cultivated land presented a fragmental trend, generally scattered around towns and rural settlements, and cropland was concentrated on both sides of the Xinshui River.

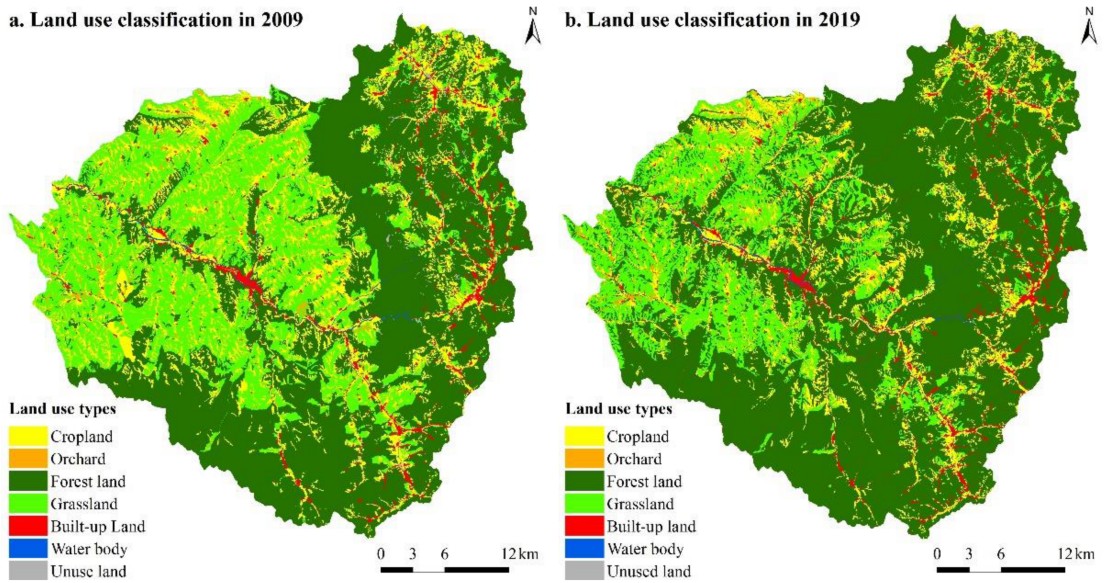

**Figure 5.** Land use classification in Pu County over the past decade.

Six typical types of land use change were further analyzed (Figure 6). The conversion of grassland to forest land was the main land use change type in Pu County over the study period, which mainly occurred in the central and northern parts with gully regions of low mountains, including the townships of Hongdao, Pucheng, southwest of Shanzhong and west of Xueguan. In these regions, the grassland was converted to forest land. The cultivated land which is not suitable for continuous planting on the steep slopes was converted to woodland for preventing soil and water loss. The conversion of forest land to built-up land mainly occurred in the suburbs of cities with higher altitudes or better geographical conditions in Pu County. Rapid urban expansion has led to a large amount of farmland loss. To ensure food security, Puxian county has actively implemented a number of land consolidation projects and high standard farmland construction projects to increase arable land area. Its high-standard farmland construction projects in Pu County were mainly distributed in Qiaojiawan, northeast Kecheng, northern Guxian, and west of Guxian Town.

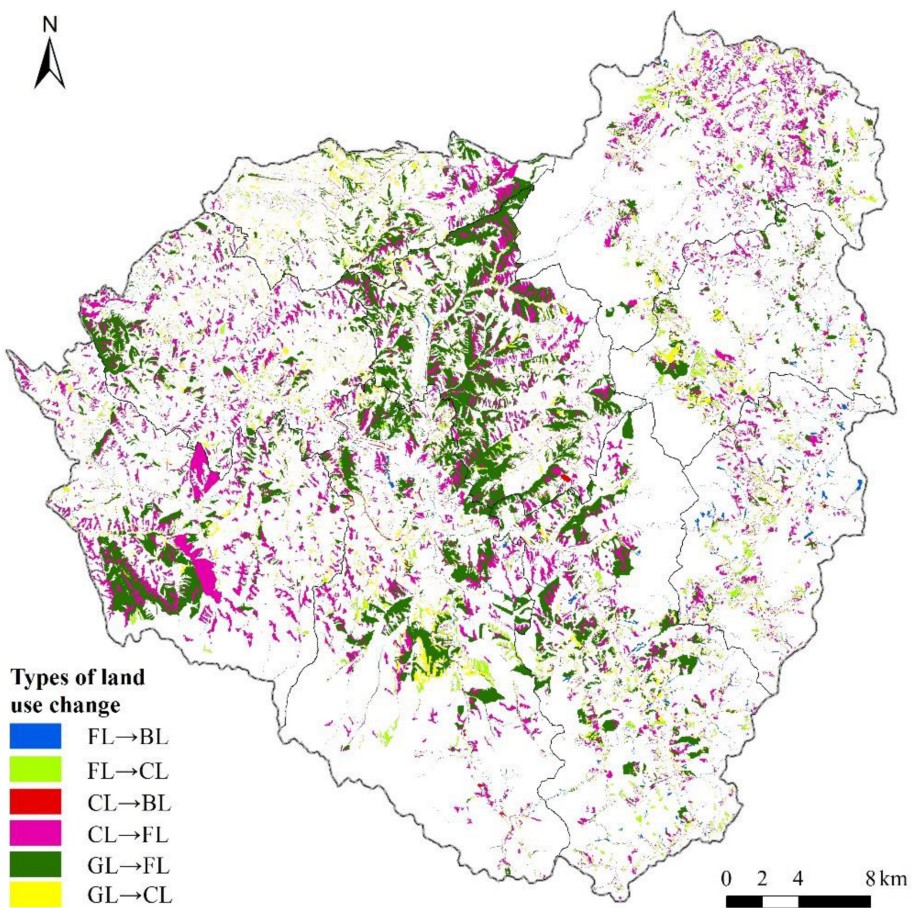

**Figure 6.** Changes in six main land use types in Pu County during 2009–2019. Notes: CL, OR, FL, GL, BL, WB, and UL represent cropland, orchard, forest land, grassland, built-up land, water bodies, and unused land, respectively.

### 3.2. Driving Forces of Land Use Transitions

Figure 7 shows the spatial distribution characteristics of nine potential factors driving land use transitions (Figure 7). Results demonstrate that topography of Pu County has obvious spatial heterogeneity, and its terrain is high in the east and low in the west (Figure 7a). The spatial differentiation of slopes is similar to that of relief degree; both indices revealed that villages in the southwest part of Pu County presented higher levels (Figure 7b,c). Due to river regulations and road construction, the road density in most villages in Pu County has increased significantly from 2009 to 2019, especially in its eastern

and central areas, and the linear distances from villages to rivers have reduced, especially in the border regions of the county (Figure 7d,e). It is obvious that spatial distribution of linear distances from villages to county forms a concentric circle radiating around the county center, and the spatial distribution of distances to towns is also shown as spreading from all the townships to the surrounding villages. Regarding population and economic growth between 2009 and 2019, villages with relatively rapid economic and population growth are mainly distributed in the northwestern region of the county (Figure 7h,i).

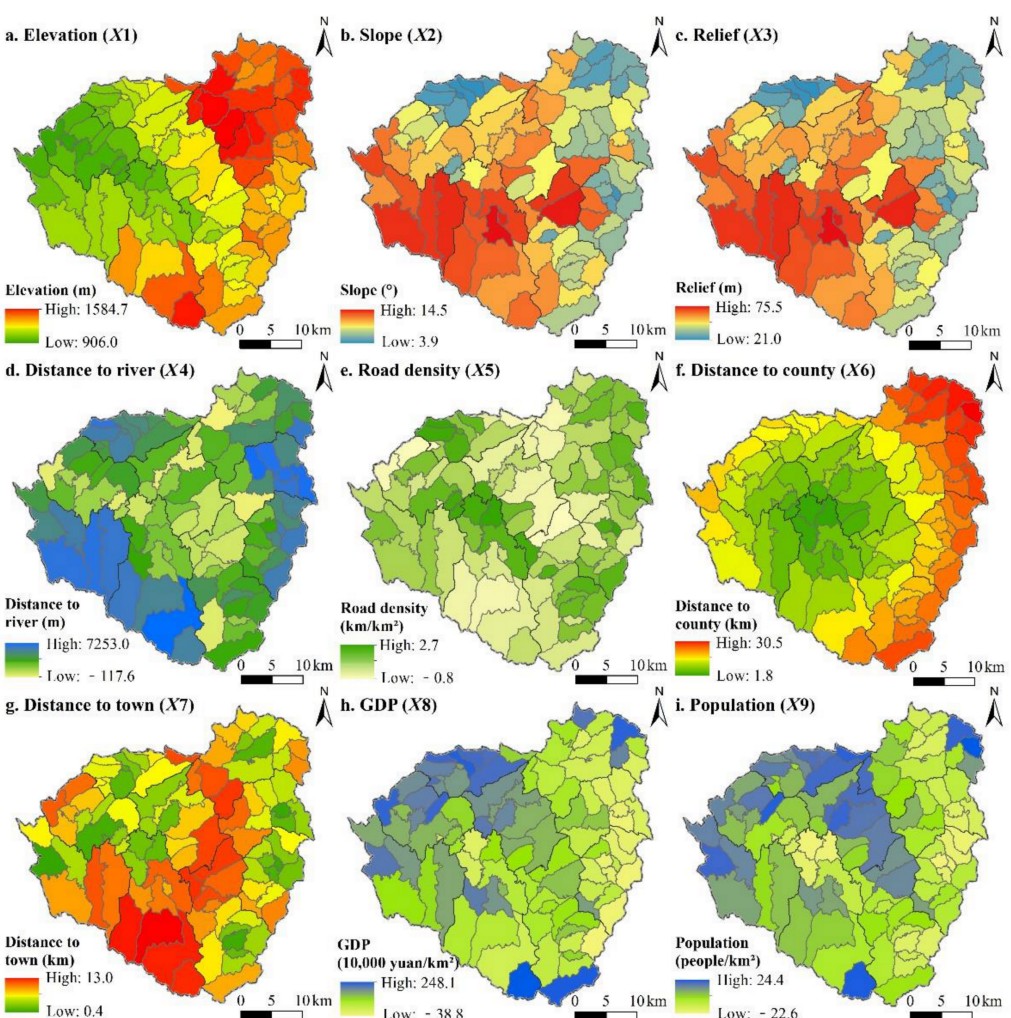

**Figure 7.** Spatial distribution of main factors driving land use transition in Pu County.

The results of factor detection are shown in Figure 8. The results demonstrated that, except for the increase in road density (X5) and distance to town (X7), most factors showed significant correlations on six types of land use transitions. All land use transitions were significantly affected by at least one factor. Specifically, two factors, i.e., distance to county (X6) and the increase in population density (X9), were identified as the main contributors to the transition of four types of land use transitions. The elevation (X1) showed significant correlations on three types of land use transitions. In addition, slope (X2) and relief degree (X3) were significantly correlated with the transition from grassland and cropland to forest land, and the distance to rivers (X4) and GDP (X8) had a significant effect on the conversion from forest land to built-up land. These results indicate that land use transformation in the Loess Plateau is driven by multiple factors.

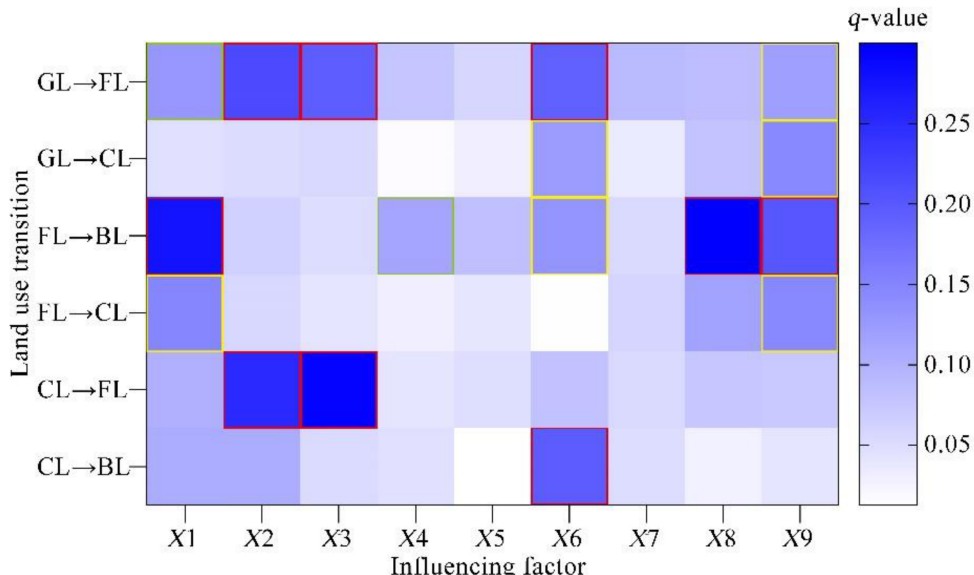

**Figure 8.** Driving forces of land use transitions in Pu County during 2009–2019. Notes: CL, OR, FL, GL, BL, WB, and UL represent cropland, orchard, forest land, grassland, built-up land, water bodies, and unused land, respectively. *X1–X9* represent elevation, slope, relief, distance to river, road density, distance to county, distance to town, GDP, and population of a village, respectively. Red frame: significance at 1%; yellow frame: significance at 5%; green frame: significance at 10%.

Specifically, firstly, five factors, i.e., slope (X2), relief degree (X3), distance to town (X6), population density (X9), and altitude (X1) were identified as the main driving factors for the transition from grassland to built-up land, whose *q*-values were 0.21, 0.19, 0.19, 0.12, and 0.13, respectively. The conversion of grassland to forest land was mainly distributed in Pu County, which was significantly correlated to slope (X2), relief degree (X3) and distance to town (X6) at a 99% confidence level, population density (X9) at a 95% confidence level, and altitude (X1) at a 90% confidence level. Sparsely inhabited areas with high slopes and poor location conditions were prone to the conversion from grassland to woodland. Compared with grassland, the rational development of forest land can help prevent soil erosion and protect the environment. Additionally, policy measures, such as the GFG and ecological compensation, also prompted grassland to be converted to forest land.

Secondly, distance to town (X6) and population density (X9) were identified as the dominant driving factors for the conversion of grassland to cropland. The transition from grassland to cropland was significantly correlated with the distance from village to town (X6) and population density (X9) at a 95% confidence level, in which their *q*-values were 0.14 and 0.10, respectively. This type of land use transition mainly occurred in the suburbs and densely populated areas. Industrialization and urbanization have led to rapid built-up area expansion, and high-quality cultivated land reduced sharply around the town. Against this background, urban residents have reclaimed some grassland for cultivating food crops on the outskirts of the county for the convenience of farming. Furthermore, a large amount of grassland in areas with low population density in the eastern part of the county was reclaimed and remediated as high-standard farmland. Thirdly, five leading factors which drove the transition of woodland to built-up land included altitude (X1), economic development (X8), population growth (X9), distance to town (X6), and the distance to rivers (X4), whose *q*-values were 0.28, 0.30, 0.20, 0.13, and 0.11, respectively. In addition, the transition had a significant correlation with the altitude (X1), GDP (X8), population density (X9), distance to town (X6) and distance to rivers (X4) at a significance level of 90% or higher. Economic growth (X8) was also identified as the driving factor, with the highest *q*-value for the conversion of forest land to built-up land. Additionally, factor detection demonstrated that altitude (X1) and population density (X9) were identified as the dominant factors for the transition from forest land to cropland, and the *q*-value of these two factors was

0.15. Based on the "Land Remediation Planning of Pu County (2016–2020)", most high-standard farmland construction projects in Pu County were carried out in the eastern part of the county, where the terrain was relatively flat, and population was relatively sparse. Therefore, the transition from forest land to cultivated land in these regions will not cause damage to the ecological environment, but alleviate the pressure caused by food security concerns. Furthermore, two factors, i.e., slope (X2) and relief level (X3), were identified as the main driving factors for the transition from cropland to forest land, whose $q$-values were 0.25 and 0.29, respectively. To prevent soil erosion, local governments have carried out the GFG programs in some gully areas with steep slopes and high surface undulations, where the ecological environment was fragile and soil erosion was severe. The southwest of the county is a relatively fragile area of ecological environment, and the implementation of the policy of returning farmland to forest is relatively strong, which promotes the transformation of large-scale cultivated land to forest land. Finally, the distance from villages to the county government station (X6) was identified as the dominant factor for the conversion from cropland to built-up land. As we know, the highest-level correlation between distance to county (X6) and the transition from cropland to built-up land was mainly due to the sharp reduction in cultivated land in the process of urban expansion.

## 4. Discussion

Multiple factors, such as natural conditions, socioeconomic development, and policies, jointly drive the rapid transformation of land use in the Loess Plateau, and the dominant driving force varies with time and region [39]. In this study, we found that the distance from villages to county was identified as the main driving factor for the conversion from cropland and forest land to built-up land, which does not disagree with previous studies [23]. From 2009 to 2019, the total area of built-up land in Pu County increased by 9.4 km$^2$, while the increase in area in Pucheng Town, i.e., the central town, was 3.4 km$^2$. Terrain conditions, such as slope and relief degree, played a dominant role in forest land expansion. Further analysis also demonstrated that population growth drove the conversion of other land use types to cropland [40]. In addition, improvements in traffic conditions were not identified as one of the main driving forces for land use transitions in this study, indicating that road construction did not occupy arable land on a large scale.

The institutional environment, such as the implementation of some engineering measures and land use policies, also plays an indispensable role in the process of land use transitions in the Loess Plateau gully regions [35,41]. In Pu County, rapid urbanization has led to the reduction in cropland (69.1 km$^2$) and expansion of built-up land (9.4 km$^2$) over the study period. The balance of cultivated land divination and the basic farmland protection system in China has greatly slowed down the loss of cultivated land in the hilly and gully areas of the Loess Plateau [3,4,42,43]. The policy of returning farmland to forests has driven the large-scale conversion of cultivated land into forest land in the Loess Plateau [44]. Previous studies have shown that the policy of returning farmland to forests has driven the large-scale conversion of cultivated land into forest land in the Loess Plateau. At the same time, land improvement engineering measures have also promoted the conversion of unused land to cultivated land [5,20].

As a relatively underdeveloped region, China's Loess Plateau region has been experiencing a far-reaching urban–rural transition development since 2009. At the same time, with the acceleration of industrialization and urbanization, the Loess Plateau is also facing an accelerated process of land use transitions, which will bring about foreseeable threatening for regional economic sustainable development and eco-environment conservation [45,46]. Therefore, in order to formulate scientific, reasonable, and sustainable land use policies, it is essential to study land use transition at the micro-scale in the Loess Plateau [47]. More research of land use transitions to reveal the effects of the interaction between biophysical environment and human activities are needed, because only in this way can we fully consider the suitability and difficulty of land use transitions and ensure that land use transitions take place within an appropriate area.

## 5. Conclusions and Policy Implications

Based on high-resolution land use data from 2009 and 2019, this study used a land use transfer matrix and the GeoDetector model to explore spatio-temporal patterns and the driving forces of land use transitions at the village level in Pu County. The results indicated that forest land, grassland, and cropland were the major land use types, and the total area of these accounted for more than 95% of the total area in Pu County. Over the study period, the land use pattern of Pu County changed significantly, characterized by the increase in orchard, woodland, construction land and water bodies, and the decrease in cultivated land, grassland, and unused land. From 2009 to 2019, land use transitions in Pu County were mainly characterized by the mutual conversion of cultivated land, woodland and grassland. Among them, about 103.3 km$^2$ and 7.5 km$^2$ of arable land were converted into forest land and built-up land, respectively, 18.4 km$^2$ and 7.7 km$^2$ of forest land were converted into arable land and built-up land, respectively, and 21.9 km$^2$ and 139.1 km$^2$ of grassland were converted into cultivated land and forest land, respectively. The conversion of grassland to forest land was the main form of land use transition in Pu County, especially in its low mountain and gully areas. The transformation of grassland to construction land was mainly distributed in the suburbs of cities and towns.

Land use transitions in Pu County were the result of the comprehensive effects of natural conditions, geographic locations, socioeconomic levels and land management systems. The driving forces in different land use transitions were distinct. Terrain, geographic location, and population growth were identified as the main driving factors for the conversion from grassland to forest land, while terrain conditions, such as slope and relief level, were the core contributors to the transition of cropland into forest land. The main driving factors for the transition from grassland to cropland were the distance to county town and population growth, and slope and population growth played a dominant role in the transition from forest land to cropland. Multi-factors, such as elevation, distance to county and rivers, economic and population growth, drove the transition of woodland to built-up land, while the distance of village to the county center was identified as the only driving factor for the conversion of cropland to built-up land.

Our findings are crucial for local policymakers to formulate future land use policies. Extreme climatic conditions and intense human activities have created an extremely fragile ecological environment in the Loess Plateau region, which principally manifests as serious soil erosion, along with vegetation degradation. Population migration to cities and economic development under the background of rapid urbanization are bound to the increasing demand for construction land and continuous expansion of built-up land, excessively occupying land space for cropland. It is thus inevitable for local governments to take measures to intensify farmland conservation, to ensure food security, and to strictly implement the balance system of requisition–compensation of farmland as well as the basic farmland protection system. Besides, since the beginning of the 21st century, vegetation coverage rate in the Loess Plateau has been greatly increased and soil erosion has been controlled to a certain extent by various engineering measures (such as slope treatment, joint management of gully and slope and watershed governance) and the GFG program. However, local governments are also facing the tradeoff between the GFG and improving farmers' livelihoods. In future, local planning-designers and policymakers need to not only curb soil erosion, but also improve land productivity, and to not only increase vegetation coverage rates but also build up a stable and high-quality land vegetation ecosystem. It is also necessary to try to take advantage of the favorable climatic conditions of climate warming and to develop an agricultural planting mode of planting two rounds per year instead of one.

**Author Contributions:** Conceptualization, Y.Z. and H.H.; software, H.H.; validation, Y.Z.; data curation, M.Q.; writing—original draft preparation, H.H.; writing—review and editing, Y.Z. and H.H.; supervision, Y.Z. and Z.Z.; All authors have read and agreed to the published version of the manuscript.

**Funding:** This study was funded by the Strategic Priority Research Program of Chinese Academy of Sciences (Grant No. XDA23070301) and the National Natural Science Foundation of China (Grant Nos. 41871183 and 41601172).

**Institutional Review Board Statement:** Not applicable.

**Informed Consent Statement:** Not applicable.

**Data Availability Statement:** The data presented in this study are available on request from the corresponding author.

**Conflicts of Interest:** The authors declare no conflict of interest.

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
