# Peer review of "Land Use Transition and Driving Forces in Chinese Loess Plateau: A Case Study from Pu County, Shanxi Province"

_land, doi:10.3390/land10010067_

Round 1

Reviewer 1 Report

Paper is well written and describes and interesting topic. The method described are not new but as a case-study type of a paper are properly used. The biggest flow of the paper is the general lack of novelty. On the plus side research, however simple, is well prepared and executed. It does fill will the journal requirements.

There are some very minor writing and visual errors that I would like to be fixed before the paper can be published.

The paper is a nice read, but since it is mostly a well performed exercise in basic statistics there is really nothing more that I can say about it.

L25 change ‘multi-factors” to ‘many factors’

L26 change ‘capable’ to ‘allowed to”

L 43 remove ‘of’

L142 – this needs a reference

Figure 6 – I am not certain what this image shows? I think it is to small

L365 change ‘factors’ to ‘factor’

Author Response

Dear Reviewers,

Thank you for your letter and thanks for the reviewers’ and editors’ comments concerning our manuscript entitled ‘Land use transition and driving forces in Chinese Loess Plateau in recent decade: A case study from Pu County, Shanxi Province’ (Manuscript ID: Land-1039421). Those comments and suggestions are very precious and helpful for us to revise and improve our manuscript. We have considered these comments carefully and made corrections accordingly, and made response and corrections which we hope it would meet the requirements to be published. Revised portions are marked using the “Track Changes” function in red color in this revised manuscript. The main corrections in this revision and the responses to the reviewer’s comments are listed as follows:

Reviewer's comments

Paper is well written and describes an interesting topic. The method described are not new but as a case-study type of a paper are properly used. The biggest flow of the paper is the general lack of novelty. On the plus side research, however simple, is well prepared and executed. It does fill will the journal requirements. There are some very minor writing and visual errors that I would like to be fixed before the paper can be published. The paper is a nice read, but since it is mostly a well performed exercise in basic statistics there is really nothing more that I can say about it.

Comment 1: L25 change ‘multi-factors’ to ‘many factors’.

Answer:

Thanks for your comments. We have adopted your suggestions and changed ‘multi-factors’ to ‘many factors’. Please see P1, L29 using the “Track Changes” function in the revised manuscript.

Comment 2: L26 change ‘capable’ to ‘allowed to”.

Answer:

Thanks a lot. We have adopted your suggestion and modified ‘capable’ to ‘allowed to’. Please see P1, L31 using the “Track Changes” function in the revised manuscript.

Comment 3: L43 remove ‘of’.

Answer:

Thank you very much. We have corrected this error and removed ‘of’. Please see P2, L49 using the “Track Changes” function in the revised manuscript.

Comment 4: L142 – this needs a reference.

Answer:

Thank you very much for your valuable comments. We have taken your advice and added some reference in the revised manuscript. Please see P4, L153 using the “Track Changes” function.

Comment 5: Figure 6 – I am not certain what this image shows? I think it is too small.

Answer:

Thanks for your valuable comments and suggestions. We agree with you. The original Figure 6 is too small, which makes it difficult for readers to understand the purpose that we want to express. Therefore, we changed Figure 6 to make it clearer and easier to understand. Please see P11 using the “Track Changes” function in the revised manuscript.

Comment 6: L365 change ‘factors’ to ‘factor’.

Answer:

Thank you for your advice. We have corrected this error and changed ‘factors’ to ‘factor’ in the revised manuscript. Please see P14, L380 using the “Track Changes” function.

Special thanks to you for your good comments and suggestions. We learned a lot from your conscientious suggestions in treating research works. We hope that the new revision is better than before. Thanks again.

Reviewer 2 Report

The research objectives are clearly defined. The methodology and results presentation have also been well-structured.

The only minor recommendation is for the authors to clearly define the difference between Cropland and Orchard.

Author Response

Dear Reviewers,

Thank you for your letter and thanks for the reviewers’ and editors’ comments concerning our manuscript entitled ‘Land use transition and driving forces in Chinese Loess Plateau in recent decade: A case study from Pu County, Shanxi Province’ (Manuscript ID: Land-1039421). Those comments and suggestions are very precious and helpful for us to revise and improve our manuscript. We have considered these comments carefully and made corrections accordingly, and made response and corrections which we hope it would meet the requirements to be published. Revised portions are marked using the “Track Changes” function in red color in this revised manuscript. The main corrections in this revision and the responses to the reviewer’s comments are listed as follows:

Reviewer's comments

The research objectives are clearly defined. The methodology and results presentation have also been well-structured.

Comment: The only minor recommendation is for the authors to clearly define the difference between Cropland and Orchard.

Answer:

Thank you very much for your valuable comments and suggestions. We have consulted some relevant literature to clearly define the difference between Cropland and Orchard. Please refer to P3, L116-117 using the “Track Changes” function in the revised manuscript.

Special thanks to you for your good comments and suggestions. We learned a lot from your conscientious suggestions in treating research works. We hope that the new revision is better than before. Thanks again.

Reviewer 3 Report

Add some more bibliography to the discussion, especially from outside China.

Author Response

Dear Reviewers,

Thank you for your letter and thanks for the reviewers’ and editors’ comments concerning our manuscript entitled ‘Land use transition and driving forces in Chinese Loess Plateau in recent decade: A case study from Pu County, Shanxi Province’ (Manuscript ID: Land-1039421). Those comments and suggestions are very precious and helpful for us to revise and improve our manuscript. We have considered these comments carefully and made corrections accordingly, and made response and corrections which we hope it would meet the requirements to be published. Revised portions are marked using the “Track Changes” function in red color in this revised manuscript. The main corrections in this revision and the responses to the reviewer’s comments are listed as follows:

Reviewer's comments

Comment: Add some more bibliography to the discussion, especially from outside China.

Answer:

Thank you very much for your valuable comments. We have consulted some relevant literature and added more bibliography to the discussion to make our argument more reasonable, especially from outside China. Please refer to P14 using the “Track Changes” function in the revised manuscript.

Special thanks to you for your good comments and suggestions. We learned a lot from your conscientious suggestions in treating research works. We hope that the new revision is better than before. Thanks again.

Reviewer 4 Report

The paper is interesting and falls within the theme of the Land Journal. I consider that the paper is well documented, the research methodology is adequately described, and the results obtained are discussed effectively.
I recommend improving the composition of the abstract. It contains too much general information. More information on the purpose and objectives, methodology and results obtained must be included.

Author Response

Dear  Reviewers,

Thank you for your letter and thanks for the reviewers’ and editors’ comments concerning our manuscript entitled ‘Land use transition and driving forces in Chinese Loess Plateau in recent decade: A case study from Pu County, Shanxi Province’ (Manuscript ID: Land-1039421). Those comments and suggestions are very precious and helpful for us to revise and improve our manuscript. We have considered these comments carefully and made corrections accordingly, and made response and corrections which we hope it would meet the requirements to be published. Revised portions are marked using the “Track Changes” function in red color in this revised manuscript. The main corrections in this revision and the responses to the reviewer’s comments are listed as follows:

Reviewer's comments

The paper is interesting and falls within the theme of the Land Journal. I consider that the paper is well documented, the research methodology is adequately described, and the results obtained are discussed effectively.

Comment: I recommend improving the composition of the abstract. It contains too much general information. More information on the purpose and objectives, methodology and results obtained must be included.

Answer:

Thank you very much for your valuable comments. What you have pointed in the review report to the abstract part of our manuscript is very useful for us to revise our manuscript. In the revised manuscript, we deleted some redundant expressions, and supplemented some contents to explain our purpose and objectives, methodology and results of our research. Please see P1 using the “Track Changes” function in the revised manuscript.

Special thanks to you for your good comments and suggestions. We learned a lot from your conscientious suggestions in treating research works.We look forward to your information about my revised manuscript.We hope that the new revision is better than before. Thanks again.
